# Effect of Nisin and Storage Temperature on Outgrowth of *Bacillus cereus* Spores in Pasteurized Liquid Whole Eggs

**DOI:** 10.3390/foods14030532

**Published:** 2025-02-06

**Authors:** Binita Kumari Goshali, Harsimran Kaur Kapoor, Govindaraj Dev Kumar, Subash Shrestha, Vijay K. Juneja, Abhinav Mishra

**Affiliations:** 1Department of Food Science & Technology, University of Georgia, Athens, GA 30602, USAhk45596@uga.edu (H.K.K.); 2Center for Food Safety, University of Georgia, Griffin, GA 30223, USA; gd03883@uga.edu; 3Cargill Protein North America, Wichita, KS 67202, USA; subash1109@hotmail.com; 4Eastern Regional Research Center, Agricultural Research Service, United States Department of Agriculture, Wyndmoor, PA 19038, USA; vijay.juneja@usda.gov

**Keywords:** *Bacillus cereus*, food safety, liquid whole eggs, predictive microbiology, Baranyi model, polynomial model

## Abstract

Pasteurization is used to ensure the safety of liquid whole eggs (LWEs) before commercial distribution; however, it is insufficient to inactivate the spore-forming bacteria *Bacillus cereus*. This study investigated the effect of nisin on the growth kinetics of *B. cereus* in LWE. Samples supplemented with 0–6.25 ppm of nisin were inoculated with a four-strain cocktail of heat-shocked *B. cereus* spores and incubated at isothermal temperatures of 15–45 °C. The Baranyi model was fitted to all *B. cereus* isothermal growth profiles, generating maximum growth rate (µ_max_) and lag phase duration (LPD). The extended Ratkowsky square root model described the temperature dependency of µ_max_. A second-order polynomial model assessed the combined effects of temperature and nisin on the LPD of *B. cereus* in LWE. A tertiary model was developed and validated using three dynamic temperature profiles. Nisin significantly extended LPD at lower temperatures, while µ_max_ remained unaffected. Samples with 6.25 ppm of nisin inhibited growth for 29 days (average) at 15 °C. Although the tertiary model accurately predicted growth rates, it underpredicted LPD. Adjusting h_0_ values for each experimental condition improved LPD prediction accuracy. The study’s findings indicate that nisin is effective in inhibiting the growth of *B. cereus* spores in LWE, lowering the risk of illness.

## 1. Introduction

Eggs are an affordable protein source in the worldwide diet [1]. Egg consumption has increased in recent decades. Liquid egg products are popular in food service and commercial manufacturing because they are easier to handle and store than shell eggs [2]. The liquid egg market was USD 5.3 billion in 2023 and is expected to reach USD 9 billion in the following decade due to urbanization and lifestyle changes [3,4]. Therefore, the egg industry must meet demand while prioritizing safe production and supply.

The liquid whole eggs (LWEs) have been homogenized, pasteurized, and packaged in an aseptic carton [5]. LWEs are vulnerable to microbial contamination because they lack their primary barrier [6]. Due to their nutritional value, eggs can naturally promote foodborne pathogen growth in optimum settings [7]. All US commercial liquid whole eggs must be pasteurized to a minimum of 140 °F for at least 3.5 min, a critical control point regulated by USDA-FSIS [8]. Pasteurization/heat treatment inactivates *Salmonella*, the main pathogenic bacteria that cause foodborne outbreaks from eggs and egg products, and preserves product quality by preventing egg protein denaturation, but it does not kill spore-forming bacteria like *Bacillus cereus* [9,10].

*Bacillus cereus*, a Gram-positive, rod-shaped, aerobic to facultative anaerobic spore-former, is prevalent in soil and vegetation and rapidly spreads to other food items [11]. *B. cereus* produces emetic and enterotoxin, which cause emetic and diarrheal sickness [12]. Martínez-Blanch and Sánchez [13] developed the methodology for the specific detection and quantification of viable *B. cereus*, tested in a liquid egg. *B. cereus* group has also been isolated from egg-laying stations, raw and pasteurized liquid whole eggs, and LWE-based baking and retail items [14]. A study isolated 78 *B. cereus* psychoactive bacteria from eggshells (7 positive samples) and pasteurized liquid whole eggs (86 positive samples) from fifty egg farms and six egg-breaking firms [15]. In another French egg-laying farm investigation, 44% of eggs were contaminated by mesophilic bacteria, with 13% *B. cereus* positive samples and 10% by psychrotrophic group bacteria with 2.3% positive samples [16,17]. Eggshells are especially susceptible to *B. cereus*, which contaminates eggs via pores or minute cracks that are hard to see, polluting the LWE pool down the processing line [18]. The spores’ adhesive properties allow them to stick to stainless steel pipelines and equipment and create biofilms [19]. Previous research found *Bacillus* spp. to be viable in marketed pasteurized LWE and lab-pasteurized goods [20]. *Bacillus weihenstephanensis*, a strain that can produce toxins at low temperatures, has also been discovered in liquid egg products [21]. Pasteurized liquid egg products stored at low temperatures pose serious safety risks [10].

LWE has a 6-day to 2-week shelf life at 4 °C after pasteurization [6,9,22]. Heat-resistant bacteria may survive the pasteurization temperature due to egg proteins’ low thermostability, and this limits the product’s shelf life [23]. Pasteurization of LWEs may heat activate spores and eliminate competition from other vegetative organisms, increasing the risk [24]. These spore-forming bacteria in the product can germinate and multiply under optimal conditions. Pasteurization is a vital safety step for LWE, but its limitations against heat-resistant *Bacillus* spp. must be acknowledged and addressed, indicating the need for additional controls in processing liquid whole eggs [25].

Bacteriocin nisin has been integrated into numerous technologies and has shown promise in inactivating infections, including spore formers [26]. Food-grade lactic acid bacteria (LAB) *Lactococcus lactis* subsp. *Lactis* generates nisin, an antibacterial [27]. Nisin slows spore germination, outgrowth, and toxin generation but does not kill them [28]. Nisin has been generally recognized as safe (GRAS) by the US Food and Drug Administration (FDA) since 1988 [29]. Industrial nisin is made by suspended-cell batch fermentation with a milk-based culture medium and commercialized as Nisaplin powder [30]. Natural substances can make LWE microbially safe. *Salmonella* and *S. aureus* growth predictive models have been developed to determine the best temperature and time for keeping and distributing liquid egg products [31,32]. Despite its prevalence and health risks, no predictive models for *B. cereus* growth in LWE exist. The effect of preservative levels on this pathogen in LWE is also inadequately studied. This study’s purpose was to create and verify a predictive model to estimate *B. cereus* development from spores in pasteurized liquid whole eggs with different nisin levels at LWE manufacturing and distribution temperatures.

## 2. Materials and Methods

### 2.1. Bacterial Strains

Four strains cocktail of *Bacillus cereus* (F 4810/72, F 4512A/87, C2, and 038-2) were used for the study. Strain *B. cereus* F 4810/72 was obtained from *Bacillus* Genetic Stock Center, Columbus, Ohio, isolated from cooked rice with a characteristic of producing emetic toxin. The remaining three strains were obtained from the Center for Food Safety, Griffin, Georgia, and could produce enterotoxin. Strains *B. cereus* F 4512A/87 and *B. cereus* C2 were isolated from pasteurized milk and pasta, respectively. Strain *B. cereus* 038-2 was isolated from infant formula. The strains were maintained at –80⁰ C as stock culture in cryobeads (Pro-Lab Diagnostics Microbanks, PL.170C/G, Thermo Fisher Scientific, Waltham, MA, USA) until use [33].

### 2.2. Spore Preparation

*B. cereus* spores were prepared as described by Juneja et al. [34] Beads of all four strains of *B. cereus* were individually inoculated into 10 mL of sterile brain heart infusion broth and incubated overnight at 37 °C to obtain an active culture. Each cell suspension (100 µL) was then surface-plated on thirty NAMS: nutrient agar (Difco, BD, Sparks, MD, USA) containing manganese sulfate (0.05 g L^−1^ MnSO_4_; Fisher Scientific, Waltham, MA, USA) agar plates. The NAMS agar plates were incubated for 10 days at 37 °C to generate spores. Spores were harvested by adding 3–4 mL of sterile distilled water to the agar plates and gently scraping the surface with sterile plastic spreaders. A pool of spores from 30 plates of each strain was collected in sterile falcon tubes (50 mL; Falcon^®^ Centrifuge Tube, Conical Bottom, Corning, NY, USA) and centrifuged at 10,000× *g* for 15 min at 4 °C (Centrifuge 5804, Eppendorf, Hamburg, Germany). Following centrifugation, the supernatant was discarded, and the cells were washed twice using 10 mL of sterile distilled water with repeated centrifugation. Finally, the cell pellets were resuspended in 15 mL sterile distilled water and stored at −20 °C until further use.

To determine spore concentration, the spores were heat-treated for 30 min at 85 °C, serially diluted in 0.1% peptone water (PW), and plated on mannitol egg yolk polymyxin (MYP) agar [35]. The plates were incubated for 18–24 h at 30 °C, and colonies were enumerated. A working spore stock of each strain was stored at 4 °C until used. A spore cocktail containing the four strains of *B. cereus* was prepared immediately before the experiment by combining an equal number of spores from each suspension. This cocktail was then heat shocked for 30 min at 85 °C to kill vegetative cells and inoculated into the samples.

### 2.3. Sample Preparation

Pasteurized liquid whole eggs packed into Parex Cartons (Tetra Pak, Cargill, Mason City, IA, USA), each containing 975 mL of the product, were shipped under refrigerated condition (EPS Foam Coolers, U-Tek Mat, Sonoco Thermosafe, Arlington Heights, IL, USA) from the Cargill egg processing facility in Mason City, IA, USA. The samples were promptly moved to a walk-in refrigerator and stored at 4 °C until the study began. The day before the experiment, the liquid egg was divided into 250 mL portions in tightly sealed 250 mL flasks (Pyrex, No.1395, Corning Inc., Conrning, NY, USA), maintaining a headspace ratio of 90:10 to replicate commercial product packaging. An aqueous solution of Nisaplin (Danisco, New Century, KS, USA) was prepared using sterile deionized water and added to the flasks accordingly to achieve 2, 4, and 6.25 ppm nisin concentration in separate containers. In the United States, European Union, Japan, Australia, and New Zealand, the maximum approved level for using Nisaplin as an antimicrobial agent in pasteurized liquid eggs is 250 ppm, which is equivalent to 6.25 ppm of nisin [36,37]. The samples were left overnight at 4 °C to ensure the complete dissolution of Nisaplin into the samples. Prior to sample preparation, 10 mL of the product from individual tetra pak was aseptically transferred to the test tube, and pH was measured using a calibrated Accumet, XL 600 pH meter (Fisher Scientific, Pittsburg, PA, USA). Liquid egg without Nisaplin was taken as control samples. Similarly, water activity was also measured using a calibrated Aqua Lab 4TE (Decagon Devices, Pullman, WA, USA) both before the start and after the end of the experiments.

### 2.4. Growth Study

Properly diluted cocktail spore suspension of *B. cereus* (3 mL) was added to a 250 mL egg sample in order to achieve an initial bacterial concentration of 2.2–2.8 log CFU/mL. Following inoculation, the flask was shaken in an arc motion for at least 20 s. The inoculated samples were then incubated in a constant temperature water bath (Thermo Scientific 18802A, Lab-Line, Aqua bath, MA, USA) maintained at 15, 25, 35, 40, and 45 °C. At each pre-determined time point, the sample was removed, and the *B. cereus* population was enumerated. The sample collection continued until the cells reached a stationary phase for modeling purposes, regardless of the egg’s shelf-life at various temperatures. The sample collection period ranged from 45 d at 15 °C to 5 d at remaining temperatures. The study was conducted in three independent replications for each combination of storage conditions and nisin concentration.

### 2.5. Enumeration of B. cereus Population

At each sampling point, the flask was removed from the water bath and wiped with 70% ethanol. It was then shaken in arc motion for 20 s. Immediately after shaking, a 2 mL sample was taken out and added to a sterile test tube containing 4 mL of 0.1% peptone water (PW). The sample flask was transferred back into the water bath as soon as the sampling was performed for future testing. The sample mixture was vortexed, followed by serial dilution in 0.1% PW. A 100 µL aliquot was taken from appropriate dilution and spread plated onto MYP agar in duplicates. The plates were incubated at 30 °C, and colonies were counted after 48 h. The detection limit of 15 CFU/mL was established. Control samples (samples without nisin) were plated alongside to compare the effects of the concentration of nisin in respective storage temperatures. The bacterial counts were recorded as CFU/mL of egg.

### 2.6. Data Analysis and Modeling

#### 2.6.1. Primary Modeling

The primary model described by Baranyi and Roberts was used in this study to determine the growth kinetic parameters of *B. cereus* in eggs [38]. The growth curves for each temperature and concentration combination were built by fitting the growth data as a function of time using the Curve Fitting tool in MATLAB (version R2023a, The MathWorks, Inc., Natick, MA, USA). The growth data were transformed into units of log_10_ CFU/mL prior to curve fitting.(1)$$y_{t}=y_{0}+\mu_{max}F\left(t\right)-ln\left(1+\frac{e^{{µ}_{max}F\left(t\right)}-1}{e^{y_{max}-y_{0}}}\right)$$(2)$$\left(t\right)=t+\frac{1}{v}ln\left(e^{-vt}+e^{-h_{0}}-e^{\left(-vt-h_{0}\right)}\right)$$
where y_t_ is the concentration of bacterial cells in log_10_CFU/mL at time t, y_0_ is the initial concentration of bacterial cells in log_10_ CFU/mL, y_max_ is the maximum bacterial concentration in log_10_ CFU/mL, µ_max_ represents the maximum specific growth rate in log_10_ CFU/h, ν represents the rise in the limiting substrate assumed to be equal to µ_max_, h_0_ represents the physiological state and is equal to the product of µ_max_ and λ, where λ is the lag phase duration in hours and is calculated from the equation:(3)$$\lambda =\frac{h_{0}}{\mu_{max}}$$

From the fitted Baranyi model, four parameters were estimated: y_0_, y_max_, µ_max_, and h_0_ for each temperature and concentration combination. The Baranyi model assumes the h_0_ of inoculum to be relatively constant regardless of subsequent growth conditions when the pre-inoculation history of the inoculum is standardized [39]. In most cases, the average value of h_0_ is calculated from the initial Baranyi fitting and fitted again into the Baranyi model as a constant to estimate the other three parameters. However, in this study, the variability in ho was significantly high due to the experiment’s diverse treatment combinations. Attempting to calculate an average and refitting the data resulted in a poor fit, further complicating the analysis.

The statistical analysis of estimates of parameters from primary modeling was analyzed using JMP^®^ Pro (version 16.0.0. SAS Institute Inc., Cary, NC, USA). The suitability of the fitted model was assessed by ensuring *p* < 0.05. After meeting this criterion, goodness of fit was determined using the coefficient of determination-R^2^ and root mean square of error-RMSE statistics [40].

#### 2.6.2. Secondary Modeling

The relationship between the primary model parameters, µ_max_ and LPD, and corresponding temperatures and nisin concentrations was analyzed using the secondary model. µ_max_ was found to be affected only by temperature from ANOVA analysis. Hence, the extended Ratkowsky square root model was used to fit the maximum specific growth rates to determine the effect of temperature on µ_max_.(4)$$\mu_{max}=(a\;\times \;(T-T_{min}{)}^{2}\;\times \;(1-\;exp(b\times \left(T-T_{max}))\right)$$
where T is the temperature (°C), T_min_ and T_max_ are the theoretical minimum and maximum growth limits (°C), respectively, and a and b are the regression coefficients.

A second-order polynomial was fitted to LPD to predict the influence of temperature and nisin concentration on *B. cereus* in LWE, as follows:(5)$$LPD=\beta_{0}+\beta_{1}\left(temperature\right)+\beta_{2}\left(concentration\right)+\beta_{3}{\left(temperature\right)}^{2}+\beta_{4}\left(temperature\times concentration\right)$$
where *β*_0_ is the intercept and *β*_1_ through *β*_4_ are the estimated values of coefficients for the main effects of the two factors, squared terms, and their interactions. This polynomial approach was selected to optimize both the fitting capability and parsimony and is suitable for a temperature range of 15–45 °C and a nisin concentration range of 0–6.25 ppm. Polynomial fitting was performed using the backward step-wise regression method using JMP^®^ Pro (version 16.0.0. SAS Institute Inc., Cary, NC, USA).

Likewise, backward step-wise regression was performed for h_0_ to analyze the effect of temperature and nisin on h_0_. Hence, a second-order polynomial equation was generated:(6)$$h_{0}=\beta_{0}+\beta_{1}({nisin)}^{2}+\beta_{2}\left(temperature\times concentration\right)$$
where β_0_ is the intercept and β_1_ through β_2_ are the estimated values of coefficients for the main effects of squared terms and their interactions.

#### 2.6.3. Tertiary Modeling

A tertiary model was developed that could predict the growth of *B. cereus* as a function of varying temperatures and concentrations of nisin. The model was developed by integrating Baranyi’s primary model and secondary models. Baranyi and Roberts [38] proposed the following two first-order differential equations to represent differential growth in the model:(7)$$\frac{\mathrm{dy}}{\mathrm{dt}}=\frac{1}{1+e^{-Q(t)}}{\text{µ}}_{\max}\left[T(t)\right]\left(1-e^{y\left(t\right)-y_{\max}}\right)$$(8)$$\frac{\mathrm{dQ}}{\mathrm{dt}}={\text{µ}}_{\max}\left[T(t)\right]$$
where y (0) = y_0_ and Q(0) = ln (Q_0_) are the initial conditions. These equations were solved using the fourth-order Runge-Kutta method with MATLAB (version R2023a, The MathWorks, Inc.). After solving the equation, microbial growth could be predicted during dynamic temperature profiles under variable nisin concentrations.

#### 2.6.4. Model Validation

The tertiary model was validated by conducting separate experiments using three non-isothermal sinusoidal time-temperature profiles combined with three different concentrations of nisin: 15–45 °C; 1 ppm nisin (12 h cycle); 15–25 °C; 3 ppm nisin (48 h cycle); 15–25 °C; 5 ppm (24 h cycle). As previously mentioned, the samples were prepared and inoculated with *B. cereus* and incubated in a programmable water bath that facilitated fluctuating temperatures in a pre-set interval period. Although there might be a lag between the product and water bath temperature, this lag was assumed to be zero in the present study, i.e., the temperature of the water bath and product was considered equal [40]. Sampling was performed at pre-determined time points, and the growth of the *B. cereus* population was determined until the cells reached the stationary phase.

Accuracy and bias factors were used to evaluate the performance of the developed tertiary model [41]:(9)$$\mathrm{Accuracy}\;\mathrm{factor}=10\frac{\sum \left|\log\left(\frac{Predicted\;}{Observed}\right)\right|}{n}$$(10)$$\mathrm{Bias}\;\mathrm{factor}=10\frac{\sum \log\left(\frac{Predicted}{Observed}\right)}{n}$$
where n indicates the sample size and predicted and observed indicate predicted values and observed values, respectively.

The acceptable prediction zone (APZ) approach was also used to validate the models with dynamic concentration-temperature profiles [42]. The prediction error was determined by taking a difference between the predicted and observed values (log CFU/mL). Fail-dangerous predictions were identified by positive PE values, whereas fail-safe predictions were indicated by negative PE values. The APZ was set between prediction errors of −1.0 and 0.5 [43].

## 3. Results and Discussion

### 3.1. pH and Water Activity

The pH of LWE samples was analyzed prior to initiating experiments and reported to be in the range of 6.55–6.71, which is relatively lower than what is generally reported: 7.3–7.8 [44] and 7.68 [45]. The reason for this reduced pH level could be attributed to the addition of citric acid to maintain the freshness and color of the egg [46]. The Agriculture Marketing Service stipulates the pH range of LWE containing citric acid to be within 6.5–6.8 [47]. These pH values are in accordance with the USDA recommendations.

At the start of the experiment, the water activity (a_w_) of the liquid whole eggs was between 0.996 and 0.998. Previous studies have shown that the a_w_ of eggs typically ranges from 0.97 to 0.99 [48,49]. By the end of the experiment, the a_w_ of the LWE remained between 0.995 and 0.998, showing no significant change. Lanciotti et al. reported that *B. cereus* can grow at a minimum a_w_ of 0.951, indicating that the higher a_w_ in LWE is conducive to the growth of *B. cereus* [50].

### 3.2. Primary Models

Figure 1 shows the growth curves of *B. cereus* in pasteurized LWE treated with 0, 2, 4, and 6.25 ppm nisin and incubated at 15, 25, 35, 40, and 45 °C, along with fitted curves generated using the Baranyi model for each replicate. While the addition of nisin could delay the outgrowth of *B. cereus* spores, it did not completely inhibit the growth. The observed values for µ_max_ and LPD on each treatment for three replications are reported in Appendix A. The primary growth model fitted the experimental growth data with R^2^ values ranging from 0.87–0.99 and root mean square error (RMSE) values ranging from 0.1–0.77.

The study demonstrated that the lag phase duration (LPD) decreases while the maximum growth rate (µ_max_) increases with higher storage temperatures up to the optimal growth temperature (~40 °C). At 15 °C, LPD in samples with 6.25 ppm of nisin extended significantly (*p* ≤ 0.05) to an average of 29 days, compared to just 25 h in samples without nisin. This 25-fold increase in LPD indicates nisin’s strong inhibitory effect at lower temperatures. Additionally, 2 and 4 ppm nisin delayed growth for an average of 166 and 330 h, surpassing control samples. At 25 °C, bacterial adaptation accelerated, reducing LPD in samples without nisin to 5.42 h. However, increasing nisin concentrations extended LPD, with the highest concentration achieving 70.87–95.42 h. This trend is consistent across the higher temperature settings of 35, 40, and 45 °C. Notably, at 35 °C and 40 °C, LPD in samples without nisin dropped to ~2 h, reflecting higher metabolic activity and suggesting these temperatures favor *B. cereus* growth, diminishing nisin’s efficacy. The enhanced sensitivity of *B. cereus* to nisin at lower temperatures may be due to changes in membrane fluidity, a key target site for nisin [51]. Nisin’s activity relies on interacting with lipid molecules in the cell membrane, leading to pore formation or hindering cell wall formation. Reduced temperatures increase the proportion of unsaturated fatty acyl chains, amplifying membrane fluidity [52]. This enhanced fluidity improves nisin’s incorporation, forming complexes with lipids that disrupt bacterial cell function and inhibit growth. Jaquette and Baeuchat [33] determined the effectiveness of nisin in controlling the growth of *B. cereus*, which was more pronounced at 8 °C and decreased in pH from 6.57 to 5.53. Shrestha and Hariram [10] reported that the use of 6.25 ppm nisin completely inhibited the growth of *Bacillus weihenstephanensis* spores (related to the *B. cereus* family) in LWE stored at 4 °C for four weeks and at 7 °C or 10 °C for subsequent nine weeks (total thirteen weeks), further supporting the enhanced activity of nisin at lower temperatures. A similar observation was made by Oshima et al. [53], who demonstrated that the use of 6 ppm nisin inhibited the growth of *B. cereus* in milk pudding (7.5% fat) for 28 days when stored at 15 °C. The stronger inhibitory effect of nisin on *B. cereus* has also been reported in other food systems, including meat products such as beef gravy, jerky [54,55], and cooked rice and milk [56]. However, at 20 °C and 30 °C, the growth started within 2 days of incubation, indicating temperature preference for growth and mesophilic trend of the strains.

The h_0_ parameter in the Baranyi equation describes the physiological state of cells during growth and is usually constant for a specific bacterial species in a given substrate [38]. Typically averaged across all temperatures for refitting other model parameters, the h_0_ value in this study showed significant variability (2.03–122.03), mainly influenced by nisin concentrations. Averaging h_0_ led to poor fits and unrealistic results, with higher nisin concentrations increasing h_0_ values, indicating a dose-dependent effect that directly impacted bacterial LPD. Figure 1 illustrates significant variability in LPD among replicates at all storage temperatures, which increased with addition and higher nisin concentrations. This variability likely happened because nisin-induced stress forces cells to adjust before resuming growth, and the adjustment rate varies widely among cells [57]. Consequently, nisin introduces substantial variability in the system, complicating experimental outcomes and potentially impacting subsequent model development.

Unlike LPD, µ_max_ was influenced solely by temperature, with different nisin concentrations having no significant effect (ANOVA, *p* = 0.52). Additionally, the variability in µ_max_ is relatively low as compared with LPD. A study conducted by Jia et al. observed a similar trend using ε-polylysine (ε-PLH) to inhibit *Listeria monocytogenes* in fish balls, where increased concentrations delayed the lag phase but did not affect the growth rate [58]. These studies suggest that once cells enter the log phase, they grow at a consistent rate regardless of nisin concentration, likely due to nisin depletion and activity loss as well as resistance development in the pathogen. *B. cereus* is known to produce the enzyme nisinase, which degrades nisin, consequently reducing the concentration to negligible levels that are not enough to affect the rate at which the growth occurs [59,60]. Moreover, modifications in cell wall and membrane phospholipid composition have also been found to confer resistance in pathogens [61].

### 3.3. Secondary Models

The effect of temperature on the maximum growth rate (µ_max_) of *B. cereus* in LWE was analyzed by fitting the extended Ratkowsky square root model to µ_max_ values from primary modeling [62]. Figure 2 visually illustrates the increase in µ_max_ with increasing temperature till it reaches the optimum of 40 °C and decreases beyond that optimal temperature range. The R^2^ and RMSE values were reported to be 0.78 and 0.19, respectively. The model estimated values of T_min_ and T_max_ for *B. cereus* in LWE as 7.27 and 49.56 °C, respectively. These values are consistent with the findings of previous investigations on *B. cereus* in various food matrices, such as cooked rice, pasta, and beans [34,63]. It is crucial to understand that T_min_ and T_max_ are theoretical temperatures beyond which growth is not anticipated and do not align with the actual extreme growth temperatures observed in the literature [64]. Generally, the observed minimum and maximum growth temperatures are frequently higher and lower, respectively, than the extrapolated values. In this study, *B. cereus* grew at both 15 and 45 °C.

Table 1 shows how temperature and nisin concentration affect *B. cereus* lag phase duration (LPD) using the quadratic model in Equation (5). The table shows that all variables considered are meaningful in linear, quadratic, and interaction models. The fitted model was very significant (*p* < 0.0001) and accurately predicted LPD variation (R^2^ = 0.73) due to nisin concentration and temperature within experimental limitations. A system with more natural variation may tolerate a higher RMSE [65]. Spore measurements of bacterial growth are more unreliable than vegetative cell measurements. Thus, spore inocula have a higher permissible RMSE than vegetative cell inocula. A Pareto graph of model parameters’ normalized effects is shown in Figure 3. The figure shows that concentration inhibited most, followed by temperature, their interaction, and the quadratic effect of temperature. Figure 4 shows the LPD response surface plot, showing how temperature and nisin concentration affect *B. cereus* growth. After peaking at maximum concentration and minimum temperature, the figure slopes downward as temperature and concentration increase. When 6.25 ppm nisin is used at these temperatures, LPD levels off, suggesting consistency. This is also seen at lower nisin concentrations. In Equation (11), a quadratic polynomial model (R^2^ = 0.70) was employed to describe how temperature and nisin concentrations affect the cell’s physiological state, h_0_.(11)$$h_{0}=4.24+1.17\;{\left(Nisin\right)}^{2}+0.0978\;Temperature\times Nisin$$

### 3.4. Model Validation

Table 2 shows that the tertiary model established in this study was validated through experiments with three sinusoidal temperature profiles containing varied nisin doses. The constructed model accurately forecasted growth rate but underpredicted LPD (Figure 5), demonstrating that the observed lag phase is consistently longer than expected values across research replications and making the approach “fail-safe”. Food safety models may tolerate some prediction errors leaning towards a fail-safe direction, but extremely fail-safe models can destroy safe goods, resulting in severe economic losses [66]. Microbial growth models can accurately forecast growth rates, although LPD projections are generally unreliable [67,68,69]. Current environmental conditions are used to mimic LPD, but many underlying causes affect it and cannot be accurately accounted for and controlled in models. These factors include cell physiological history, prior stress, inoculum size, preculture conditions, environmental change rate and magnitude, and strain variability [70,71,72,73,74].

This study also included varied *B. cereus* cell pre- and post-incubation temperatures. When pre- and post-incubation temperatures were matched, a previous study found the shortest LPD [75]. Unintentionally, the validation study repeated inoculum sizes diverged by 0.5 log CFU/mL. This likely caused differing LPDs for validation study replicates. The inoculum size for replicate 2 exceeded 2.7 log CFU/mL for sinusoidal profile 1 (Table 2), compared to 2.2 in replicate 1. In replicate 2, LPD dropped significantly and entered the “fail-dangerous” side of the model forecast. In the lag phase of *L. innocua*, a 0.25 log CFU/mL increase in inoculum size decreased LPD by 23% [76]. Quorum sensing and cell signaling may explain inoculum size-induced LPD variance. Quorum sensing uses autoinducers to regulate metabolic and behavioral processes in a colony of bacteria [77]. Increased population density increases autoinducer concentration, which can cause coordinated bacterial behavior, such as faster metabolic activity and rapid exponential phase progression [78]. Quorum sensing also improves nutrient access, activates defensive reactions against competing cells, and allows cells to change morphologically to survive in hostile, growth-inhibitive environments like antimicrobials [79]. Thus, coordination can affect lag phase length. Zhao et al. [80] found signaling molecules in *C. botulinum* quorum sensing. The study suggests spores demonstrate dependency on each other and also briefly explains inocula size effects observed in challenge studies.

Antimicrobial demand and efficacy can vary with microbial inoculum. A particular dose of nisin inhibits cell proliferation. Higher inoculum concentrations require higher nisin concentrations to target all cells and sustain antibacterial activity. However, nisin is abundant enough to kill or inhibit cells at lower doses, lengthening the lag period compared to scenarios with high bacterial loads utilizing the same concentration. Virto et al. estimated the influence of initial microbial concentration on *Escherichia coli* chlorine resistance and found a similar trend [81]. Before inoculation in the sample, the spore inoculum was heat treated, which presumably stressed and changed the cells’ physiological condition, increasing LPD variability. Polynomial models describing heat injury [82] and preservation treatments like NaCl [83] and nitrite [84] in food matrix to control *Salmonella* and *L. monocytogenes* reported differences in LPD, but pH and water activity also affected this. Polynomial models often underestimate LPD by linking growth parameters to environmental factors [85]. In a 1995 review of 755 prediction examples from seven response surface modeling articles, Delignette-Muller et al. found 468 polynomial and 287 square root models [86]. Polynomial models explaining LPD had high prediction errors (40.3%) and mean errors of 25%. A study conducted by Oscar examined 16 models from nine response surface modeling articles and found 18.5–74.8% prediction errors [87]. Peña and De Massaguer used a quadratic polynomial model to assess *Alicyclobacillus acidoterrestris* adaptation time in orange juice, including pH, temperature, soluble solids, and nisin [88]. Some treatment combinations had poor adjustment, as model projections and experimental values differed by 28%. Adding environmental elements to polynomial models makes prediction more complicated, making model validation harder and predictions erroneous [72]. Validation supports LPD variability in primary modeling.

The discrepancy in the inaccurate prediction of LPD was addressed using a tailored value of h_0_, calculated using Equation (11). The adjusted h_0_ (Table 2) predicted the LPD more accurately by matching the model assumptions to the physiological conditions of the cells in each experimental run (Figure 5), thereby reducing the variance between predicted and experimental LPD with acceptable bias and accuracy factor, as displayed in Table 2. Predictions after adjustment of h_0_ were compared with observed values, and APZ analysis was performed (Figure 6). Out of 108 predictions, 90 fell within APZ (−1 to 0.5 log CFU/mL). This adjustment approach reflects a ‘dynamic’ modeling approach, in which input parameters (h_0_ values) are tuned based on empirical evidence from experiments. Koseki and Isobe also used a similar method of using adjusted α_0_, a geometric mean value for the physiological state parameter, to fit the observed data [89]. The author emphasizes that this process is not prediction but only fitting. Model validation and adjustment using this iterative process turned out to be critical for improving the predictive accuracy of models in microbial kinetics.

## 4. Conclusions

The main outcome of this study was that nisin concentrations did not impact the maximum growth rate, but they did extend the lag phase at each storage temperature, which was significant at lower temperatures and higher concentrations. The tertiary model that predicted growth behavior underpredicted LPD, making it fail-safe yet accurate for growth rate. LPD varied greatly between repetitions, showing the impact of biological variety and experimental circumstances. Adjusting h_0_ for each experimental setting greatly improved the model’s accuracy, minimizing LPD discrepancies. The bias and accuracy factors after h_0_ modification were 0.89–0.99 and 1.13–1.2. Future studies should focus on refining predictive models by incorporating more dynamic and adapted approaches to accommodate the physiological state variations within bacterial populations. This could further enhance the accuracy and applicability of LPD predictions under diverse storage and processing conditions. Additionally, future research could also investigate the benefits of reducing pH to enhance nisin action and extend the lag period. However, LWE pH modification should be tested for sensory characteristics to confirm that lower pH does not alter egg organoleptic qualities. Investigating the interaction of nisin with other food matrix components, as well as its synergistic effects with other antimicrobials, represents an important avenue for research. Such advancements could pave the way for broader applications of nisin in ensuring food safety and extending the shelf life of perishable food products.

## Figures and Tables

**Figure 1 foods-14-00532-f001:**
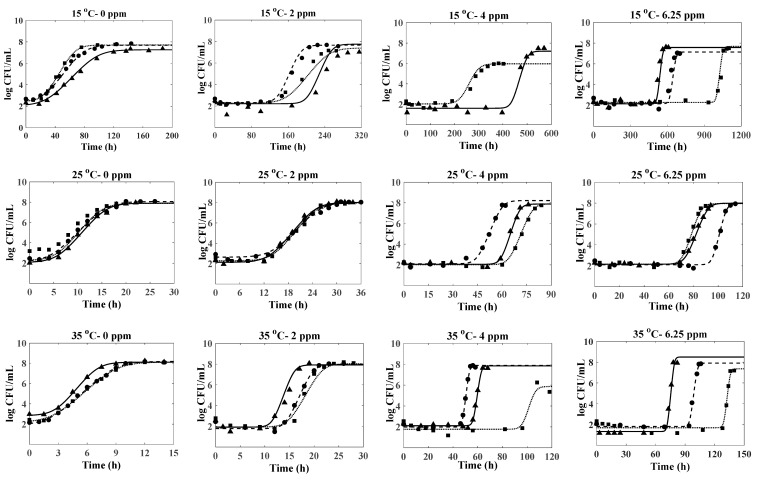
Growth of *B. cereus* (log CFU/mL) in pasteurized liquid whole eggs (LWEs) supplemented with different nisin concentrations (0, 2, 4, 6.25 ppm), stored at different temperatures (15, 25, 35, 40, 45 °C) and fitted Baranyi model. Growth data: ●—Rep 1, ■—Rep 2, ▲—Rep 3. Fitted line: − − − −—Rep 1, ∙∙∙∙∙—Rep 2, ⸺⸺—Rep 3.

**Figure 2 foods-14-00532-f002:**
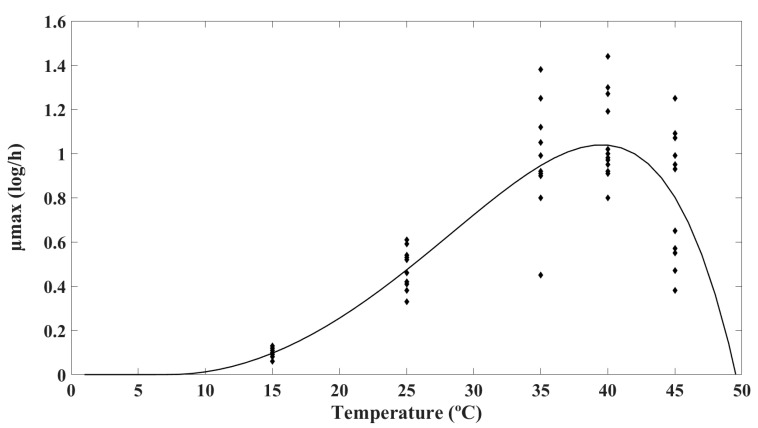
Observed values with extended Ratkowsky model. In the graph, µ_max_ is maximum growth rate (log/h), ♦ represents observed values, and the curve line represents the prediction generated by the model.

**Figure 3 foods-14-00532-f003:**
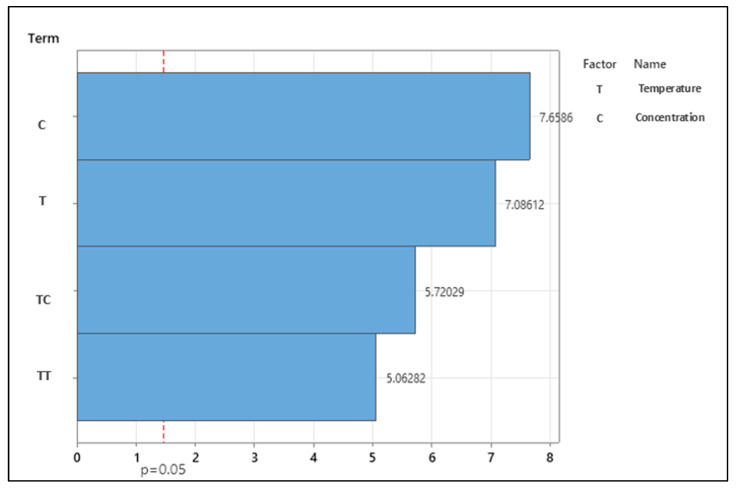
Pareto graph to evaluate the effect of parameters on lag phase duration (LPD) of *B. cereus* in pasteurized liquid whole eggs (LWEs).

**Figure 4 foods-14-00532-f004:**
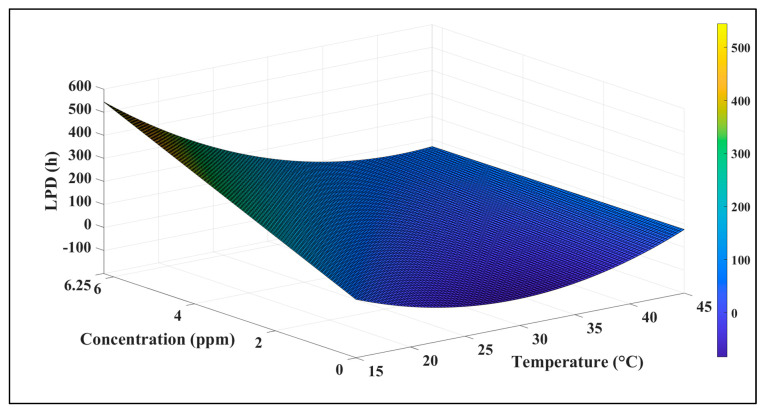
Surface plot of lag phase duration (LPD) affected by temperature and concentration of nisin.

**Figure 5 foods-14-00532-f005:**
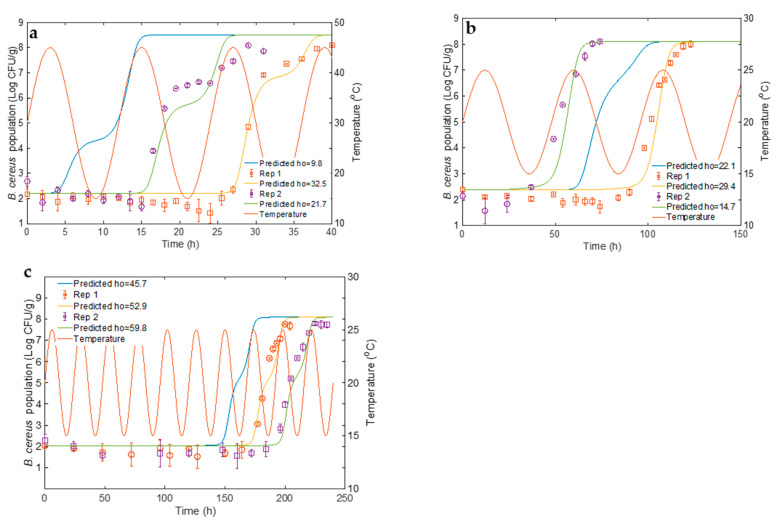
The comparison of the predictive curve and fitted curve of *B. cereus* in liquid whole eggs (LWEs) at different time-concentration-varying temperature profiles: (**a**) 15–45 °C (12 h/cycle); (**b**) 15–25 °C (48 h/cycle); (**c**) 15–25 °C (24 h/cycle).

**Figure 6 foods-14-00532-f006:**
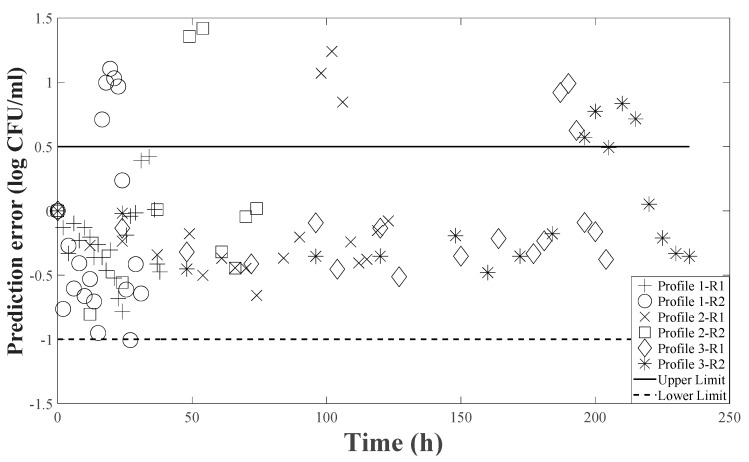
Acceptable prediction zone (APZ) analysis: data for time-concentration-temperature profiles for *B. cereus* for non-isothermal temperature profiles of 15 to 45 °C and 15 to 25 °C.

**Table 1 foods-14-00532-t001:** Relevant regression coefficients of parameters and goodness of fitness statistics of the polynomial model for lag phase duration (LPD). R^2^ is the coefficient of determination, RMSE is toot mean square error.

Summary of fit	
Probability > F	0.0001
R^2^	0.73
Adjusted R^2^	0.71
RMSE	0.89
LPD Regression	Coefficients
Parameters	
Intercept	488.81
Temperature	38.87
Nisin	118.25
Temperature^2^	0.65
Temperature X Nisin	2.65

**Table 2 foods-14-00532-t002:** Validation trials of polynomial model for lag phase duration (LPD) of *B. cereus* in pasteurized liquid whole eggs (LWEs) with adjusted physiological parameter, h_0_.

Sinusoidal Profiles	Temperature	Conc. (ppm)	Replication	h_0_	Bias Factor	Accuracy Factor
1	15–45 °C (12 h interval)	1	1	32.5	0.89	1.13
			2	21.7	0.92	1.20
2	15–25 °C (48 h interval)	2	1	29.4	0.93	1.15
			2	14.7	0.99	1.16
3	15–25 °C (24 h interval)	3	1	52.9	0.93	1.12
			2	59.8	0.97	1.13

## Data Availability

The original contributions presented in this study are included in the article. Further inquiries can be directed to the corresponding author.

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
