# Peer review of "Effect of Nisin and Storage Temperature on Outgrowth of Bacillus cereus Spores in Pasteurized Liquid Whole Eggs"

_foods, 2025, doi:10.3390/foods14030532_

Round 1
Reviewer 1 Report
Comments and Suggestions for Authors
The authors proposed a manuscript titled “Effect of Nisin and Storage Temperature on Outgrowth of Ba-2cillus cereusSpores in Pasteurized Liquid Whole Eggs”. According to them, the “study aimed to explore the effect of nisin on the growth kinetics of Bacillus cereus in liquid whole eggs.” The authors mentioned that “The statistical analysis showed that while the LPD of B. cereuswas affected by both storage temperature and nisin concentration, μmax was unaffected by nisin concentrations. Growth was considerably inhibited for 29 days on average in the sample containing 6.25 ppm at 15 °C.”
Unexpectedly, the authors did not consider the state of the art of their research subject and did not cite important previously published works:
Combined effects of pH, nisin, and temperature on growth and survival of psychrotrophic Bacillus cereus
JAQUETTE, C. B. ANDBEUCHAT, L. R., Journal of Food Protection 61(5), , 563-570 1998https://doi.org/10.4315/0362-028X-61.5.563
Detection and quantification of viable Bacillus cereus in food by RT-qPCR
Martínez-Blanch, J.F., Sánchez, G., Garay, E. et al., Eur Food Res Technol 232, 951–955, 2011. https://doi.org/10.1007/s00217-011-1465-1
So, the referee believes that the work must be reconsidered and reformulated to allow readers to understand the scientific novelty the authors intended to incorporate in their work.
Author Response
The authors proposed a manuscript titled “Effect of Nisin and Storage Temperature on Outgrowth of Bacillus cereus spores in Pasteurized Liquid Whole Eggs”. According to them, the “study aimed to explore the effect of nisin on the growth kinetics of Bacillus cereus in liquid whole eggs.” The authors mentioned that “The statistical analysis showed that while the LPD of B. cereus was affected by both storage temperature and nisin concentration, μmax was unaffected by nisin concentrations. Growth was considerably inhibited for 29 days on average in the sample containing 6.25 ppm at 15 °C.”
Unexpectedly, the authors did not consider the state of the art of their research subject and did not cite important previously published works:
Combined effects of pH, nisin, and temperature on growth and survival of psychrotrophic Bacillus cereus
JAQUETTE, C. B. ANDBEUCHAT, L. R., Journal of Food Protection 61(5), 563-570 1998 https://doi.org/10.4315/0362-028X-61.5.563
Detection and quantification of viable Bacillus cereus in food by RT-qPCR
Martínez-Blanch, J.F., Sánchez, G., Garay, E. et al., Eur Food Res Technol 232, 951–955, 2011. https://doi.org/10.1007/s00217-011-1465-1
So, the referee believes that the work must be reconsidered and reformulated to allow readers to understand the scientific novelty the authors intended to incorporate in their work.
AU: We thank the reviewer for this suggestion. We have included the reference to the suggested studies in the manuscript. (Page 2, Lines 66 - 68; Page 7, Lines 313 – 315)
Reviewer 2 Report
Comments and Suggestions for Authors
1. What is the pH and water activity of the liquid whole egg samples used in the experiment?
2. Are the four B. cereus strains used in the experiment representative of the diversity of the species?
3. Did the authors encounter any challenges in fitting the data using the Baranyi model, such as variability in the data? The article mentions that there is a lot of variability in the LPD, did this variability affect the predictive ability of the model?
4. Nisin was found to have a stronger inhibitory effect on B. cereus at lower temperatures in the study, is this consistent with the performance of Nisin in other food systems?
5. Why was the inhibitory effect of Nisin on B. cereus attenuated at 35°C and 40°C?
6. Did the authors consider all possible scenarios for different combinations of Nisin concentration and temperature during the model validation phase? Is the approach of adjusting h0 values to improve the accuracy of LPD prediction generally applicable to other types of micro-organisms and food systems?
7. Do the authors plan to investigate the interactions of Nisin with other food components and how these interactions affect the growth of B. cereus?
8. How will other environmental factors that may affect the growth of B. cereus, such as pH and water activity, be controlled during the experiment?
Author Response
- What is the pH and water activity of the liquid whole egg samples used in the experiment?
AU: The pH was in the range of 6.55-6.71 and water activity was between 0.996-0.998 (Page 6, lines 273 – 274; 279 – 280)
- Are the four B. cereus strains used in the experiment representative of the diversity of the species?
AU: Yes, the four strains used in the experiment represent diversity within the species. We used both emetic and enterotoxin producing strain types, which are common concerns in food safety. In addition, the strains also came from different food matrices. (Page 2, Lines 106 - 112)
- Did the authors encounter any challenges in fitting the data using the Baranyi model, such as variability in the data? The article mentions that there is a lot of variability in the LPD, did this variability affect the predictive ability of the model?
AU: Yes, the variability in the LPD across the replicates stemmed from diverse treatment combinations of nisin concentrations and storage temperatures. The Baranyi model fitted the experimental growth data well with R2 values ranging from 0.87-0.99 and root mean square error (RMSE) values ranging from 0.1-0.77. However, the physiological parameter, h0 in the Baranyi equation exhibited significant variability, which affected the prediction of LPD. We were able to address the variability by adjusting h0​ values for each experimental setup using equation 11 in the manuscript. This adjustment accounted for the observed variability and ensured that the predictive ability of the model remained robust, while maintaining a fail-safe approach crucial for food safety applications.
- Nisin was found to have a stronger inhibitory effect on B. cereus at lower temperatures in the study, is this consistent with the performance of Nisin in other food systems?
AU: The stronger inhibitory effect of nisin on B. cereus at lower temperatures observed in our study aligns with its performance in other food systems. Prior studies have demonstrated similar temperature-dependent efficacy, likely due to changes in bacterial membrane fluidity at lower temperatures, which enhance nisin’s ability to interact with and disrupt the cell membrane. We have added more references to the text, comparing the results from the study with the available literature which have reported pronounced effect of nisin at lower temperatures in liquid whole eggs and other food systems including meat products such as beef gravy, jerky, and cooked rice, and in milk. (Page 7, Lines 315 - 323)
- Why was the inhibitory effect of Nisin on B. cereus attenuated at 35°C and 40°C?
AU: B. cereus is a mesophilic bacterium with an optimal growth range between 30°C and 40°C. At these temperatures, its metabolic activity is heightened, and it can grow more rapidly, reducing the lag phase even in the presence of nisin.
- Did the authors consider all possible scenarios for different combinations of Nisin concentration and temperature during the model validation phase? Is the approach of adjusting h0 values to improve the accuracy of LPD prediction generally applicable to other types of micro-organisms and food systems?
AU: During the validation phase, we considered three sinusoidal non-isothermal temperature profiles combined with three different nisin concentrations (1 ppm, 3 ppm, and 5 ppm) to test the developed tertiary model. These profiles represented dynamic temperature fluctuations that are likely encountered during the processing, storage, and distribution of liquid whole eggs. While it is impossible to test every possible combination of temperature and nisin concentration, the selected scenarios encompassed a realistic range of conditions within the experimental design. This ensured that the model could be validated effectively for practical applications.
Yes, the approach of adjusting h0​ values is potentially applicable to other microorganisms and food systems, especially in cases where significant variability is observed in LPD. Tailoring h0​ values based on specific experimental conditions can account for biological variability, and environmental factors. This dynamic modeling approach has been utilized in previous studies, such as by Koseki and Isobe (2012), to improve the predictive accuracy of microbial growth models. (Page 14, Lines 600-602)
- Do the authors plan to investigate the interactions of Nisin with other food components and how these interactions affect the growth of B. cereus?
AU: In the future, we aim to explore these interactions in more detail, particularly in relation to food components such as lipids, proteins, and additives commonly present in liquid egg products and other matrices. Understanding these dynamics will provide deeper insights into optimizing nisin’s application for food safety.
- How will other environmental factors that may affect the growth of B. cereus, such as pH and water activity, be controlled during the experiment?
AU: Regular monitoring of pH during the experiments is essential to ensure that pH levels remain within a specific range. To prevent fluctuations in water activity (aw), all samples were tightly sealed in flasks, maintaining a headspace ratio of 90:10 to replicate commercial product packaging to avoid moisture loss. Additionally, temperature-controlled incubation minimize evaporation and condensation effects, ensuring stability throughout the experiment.
Reviewer 3 Report
Comments and Suggestions for Authors
Dear Editor and Authors,
I send you my comments about the Article “Effect of nisin and storage temperature on outgrowth of bacillus cereus spores in pasteurized liquid whole eggs”.
The scope of the Article, as reported in the aim, was to study the effect of nisin on the growth kinetics of Bacillus cereus in liquid whole eggs.
In my opinion, although in the present version the Article is well structured, it is well written and its topic result original, I would like to suggest to the Authors some minor improvement that I reported below.
The introduction is well written, but to better explain the relevance of this topic I would like to suggest to the authors to improve the comparison with others articles already reported in the references.
Moreover, in the paragraph of materials and methods should be better highlighted the number of trials performed and the number of replicates of analysis performed in each trial.
The results is well presented and they are well discussed, also in comparison to the data reported in the literature.
However, to facilitate the comprehension of the concepts reported in the figures and the data showed in the tables their captions should be improved, to this aim I would like to suggest to the Authors to better explain in the caption the concept expressed in the figures and the data showed in the tables.
Finally, he conclusions is adequate to the results showed and they satisfy the aim of the Article.
Nevertheless, I would like to suggest to the Authors to stress the relevance of the findings reported in literature, their impact and the future prospective.
Very well done.
Best regards
Author Response
I send you my comments about the Article “Effect of nisin and storage temperature on outgrowth of bacillus cereus spores in pasteurized liquid whole eggs”. The scope of the Article, as reported in the aim, was to study the effect of nisin on the growth kinetics of Bacillus cereus in liquid whole eggs. In my opinion, although in the present version the Article is well structured, it is well written and its topic result original, I would like to suggest to the Authors some minor improvement that I reported below.
The introduction is well written, but to better explain the relevance of this topic I would like to suggest to the authors to improve the comparison with others articles already reported in the references.
AU: We thank the reviewer for their valuable feedback and positive evaluation of the manuscript. In response to the suggestion to improve the comparison with other articles in the introduction, we have added more references of the available literature to the section of results and discussion, comparing the effects of nisin in other food matrices and stress on the effectiveness of nisin in controlling growth of B. cereus. (Page 7, Lines 313 – 315; 321 – 323)
Moreover, in the paragraph of materials and methods should be better highlighted the number of trials performed and the number of replicates of analysis performed in each trial.
AU: We thank the reviewer for the suggestion. As mentioned in the manuscript, the growth study was conducted in three independent replications for each combination of storage conditions and nisin concentration. (Page 4, Lines 171 – 172)
The results is well presented and they are well discussed, also in comparison to the data reported in the literature. However, to facilitate the comprehension of the concepts reported in the figures and the data showed in the tables their captions should be improved, to this aim I would like to suggest to the Authors to better explain in the caption the concept expressed in the figures and the data showed in the tables.
AU: We thank the reviewer for the positive comment and suggestion. Captions for all the figures and tables have been improved with added detail for easy and improved understanding of the reader.
Finally, the conclusions is adequate to the results showed and they satisfy the aim of the Article. Nevertheless, I would like to suggest to the Authors to stress the relevance of the findings reported in literature, their impact and the future prospective.
AU: We thank the reviewer for the suggestion. We revised the conclusion section by adding a few sentences about the relevance of the findings, their impact, and future prospects (Page 14, Lines 549 – 560)
Reviewer 4 Report
Comments and Suggestions for Authors
The article entitled “Effect of nisin and storage temperature on the growth of Bacillus cereus spores in pasteurized liquid whole eggs” presents a model to estimate the growth and resistance of B. cereus after treatment with nisin. The work is well-designed and the results are presented correctly. The presentation of the data is of high quality, as is the discussion. I suggest some changes to the Abstract and Introduction, which can be found in the attached file. After the necessary corrections, the article may be accepted for presenting scientific merit, originality and importance to the area of ​​food safety.

Author Response
The article entitled “Effect of nisin and storage temperature on the growth of Bacillus cereus spores in pasteurized liquid whole eggs” presents a model to estimate the growth and resistance of B. cereus after treatment with nisin. The work is well-designed and the results are presented correctly. The presentation of the data is of high quality, as is the discussion. I suggest some changes to the Abstract and Introduction, which can be found in the attached file. After the necessary corrections, the article may be accepted for presenting scientific merit, originality and importance to the area of ​​food safety.
AU: We thank the reviewer for the positive comment and suggestion.
Line 13- 14: What is the importance of the study? Please add a short introduction before the objectives.
AU: We thank the reviewer for and suggestion. We have added the short introduction (Page 1, Lines 13 – 14)
Line 12: “B. cereus”. Italic
AU: Change has been made, as suggested by the reviewer.
Line 14 – Line 27: The authors need to restructure the abstract. It begins with the objectives, without briefly presenting the central theme of the study. Later, the methodology is superficial. Finally, the conclusion does not present the information for what was found in the study.
AU: We thank the reviewer for and suggestion. We have revised the abstract. (Page 1, Lines 13 – 26)
Line 14: What kind of the samples?
Please, provide more information about the methodology. In the abstract this information is very superficial.
AU: We thank the reviewer for and suggestion. It refers to liquid whole eggs (LWE) and we have revised the abstract. (Page 1, Lines 13 – 26)
Line 15: Italic... Please, check in the all text.
AU: Change has been made, as suggested by the reviewer.
Line 47 – 50: Strikethorugh
AU: We thank the reviewer for the suggestion. We have revised the lines without deleting the basic information about B. cereus. (Page 2, Lines 61 – 65)
Line 52 – 53: Are there data on cases/outbreaks of foodborne illnesses relating eggs and B. cereus? If so, add them here.
AU: Data linking B. cereus to outbreaks involving egg products are limited. Nonetheless, the potential for contamination exists, especially if eggs or egg products are improperly handled or stored, providing favorable conditions for bacterial growth (Page 2, Lines 68 – 81)
Round 2
Reviewer 2 Report
Comments and Suggestions for Authors
ok